# Classification and Recognition of Building Appearance Based on Optimized Gradient-Boosted Decision Tree Algorithm

**DOI:** 10.3390/s23115353

**Published:** 2023-06-05

**Authors:** Mengting Hu, Lingxiang Guo, Jing Liu, Yuxuan Song

**Affiliations:** 1School of Civil Engineering and Architecture, Xiamen University of Technology, Xiamen 361024, China; 2The Arts and Design College Xiamen, Fuzhou University, Xiamen 361000, China; 3College of Forestry and Grassland, Jilin Agriculture University, Changchun 130118, China

**Keywords:** decision tree algorithm, building classification, k-fold cross-validation method, model cluster

## Abstract

There are high concentrations of urban spaces and increasingly complex land use types. Providing an efficient and scientific identification of building types has become a major challenge in urban architectural planning. This study used an optimized gradient-boosted decision tree algorithm to enhance a decision tree model for building classification. Through supervised classification learning, machine learning training was conducted using a business-type weighted database. We innovatively established a form database to store input items. During parameter optimization, parameters such as the number of nodes, maximum depth, and learning rate were gradually adjusted based on the performance of the verification set to achieve optimal performance on the verification set under the same conditions. Simultaneously, a k-fold cross-validation method was used to avoid overfitting. The model clusters trained in the machine learning training corresponded to various city sizes. By setting the parameters to determine the size of the area of land for a target city, the corresponding classification model could be invoked. The experimental results show that this algorithm has high accuracy in building recognition. Especially in R, S, and U-class buildings, the overall accuracy rate of recognition reaches over 94%.

## 1. Introduction

The complexity of land use in cities around the world has increased. Land use planning and classification have become an important ways to coordinate urban development and achieve economic, scientific, and rational utilization of land resources. Clarifying the various types of urban land use is imperative for planning. However, the diversification of business types caused by improvements in urban economic levels has led to the agglomeration of land use types with massive scales and high complexity in many cities. Providing an efficient and scientific identification of the types of large-scale spatial land uses has not only become a major difficulty in planning, but also represents a basis for the scientific monitoring of urban development. Therefore, the complex mixing and high-frequency changes in land use functions have led to the automation of the large-scale identification of urban land, as supplemented by manual verification.

The quantitative processing of big data, such as in intelligent maps, search engines, transportation data, remote sensing images, and business-type point of interest (POI) data, is currently a common approach to the automatic recognition of urban land. Wu et al. [1] and Li et al. [2] used remote sensing image data for Python programming and a spatial proximity analysis to accurately identify the boundaries of urban construction land at different scales. Han et al. [3] and Cao et al. [4] used bus “swiping” data to construct an urban functional area identification model for clustering the data and identifying urban functional areas with a certain degree of matching with a land use status map. Peng et al. [5] used location-based service (LBS) high-precision positioning data to extract the time-series characteristics of residential call aggregations as oriented on a plot scale. Peng proposed a method for recognizing urban land use type spectrum clustering. In addition, Rong et al. [6], Liu [7], Zhao et al. [8], Anning [9], and Zeng [10] used business-type POI big data and other open-source data to identify and evaluate urban land types using weighted, entropy-weighted, and mathematical models. Therefore, using through business-type POI data is one of the most common automated methods for identifying urban land.

At present, the accuracy and precision of automatic land use recognition require further improvement. For example, in a complex urban space, the bus-swiping data and passenger flow data are local data. There is no corresponding relationship between these data and the land type, making it difficult to ensure the accuracy of land recognition. However, relying solely on business-type POI data to identify land use types makes it impossible to accurately identify plots with mixed functions and multiple business types, especially public service plots with mixed uses such as commercial–residential and commercial–office. Nevertheless, owing to the great differences between the data classification standard of the business-type POI and the land classification of the national standard, only residential, commercial, industrial, and other urban land categories can be identified using mathematical models or entropy weight calculations. Moreover, it is impossible to further refine this recognition using these approaches; ultimately, they cannot reach the classification accuracy of the national land use standard. Therefore, it is difficult to apply them widely in actual planning.

Based on the above problems and based on the big data of business form POIs, this study aims at the type-identification problem of multiple and mixed business forms, integrates the building form database, and conducts cross-identification of land use types according to various morphological indicators of the plot from a morphological perspective. The accuracy is further improved by introducing the cross-identification of the multi-source heterogeneous big data. It exceeds the limitations of the mathematical models, weighting calculations, and other methods of analyzing the business-type POI data by adopting an artificial intelligence (AI) method based on spatial calibration and weighted transformations of business types and building spatial data. Systematic supervised classification learning and intelligent identification are conducted for cities of different types, sizes, and locations so that the identified categories are entirely consistent with the national land classification standards. Simultaneously, intelligent rules are embedded according to the dimensions of the shape indices, layout shapes, and special shapes. The precision and accuracy of the recognition are further improved through confidence labeling, research feedback, and system self-optimization, as shown in Figure 1. RAHMAN et al.’s [11] main goal was to describe the performance of different machine learning algorithms on three different spatial and multispectral satellite image classifications in rural and urban areas. They used random forests and support vector machines (SVM), and their combined strengths were applied separately to Landsat-8, Sentinel-2, and Planet images separately to assess the individual and overall class accuracy of the images. CHAIB et al. [12] proposed a new deep framework is proposed for very high-resolution (VHR) scene understanding by exploring the strengths of vision transformer (ViT) features in a simple and effective way. This pre-trained ViT model is used to extract informative features from the original VHR image scene, where the transformer–encoder layers are used to generate the feature descriptors of the input images. Xiong et al. [13] proposed a novel single-stage 3D object detection network based on density clustering and graph neural networks. Which utilizes density clustering ball queries to partition the point cloud space and exploits local and global relationships by graph neural networks. Zhan et al. [14] describe how DL-based methods have been utilized for subsurface sedimentary structure identification from the viewpoint of different identification approaches (direct and data assimilation-based modeling), and the differences between DL-based and traditional methods are discussed.

In the related research on urban architecture and space. Chen et al. [15] analyzed the relationship between the population distribution and the underground space use of the central city of Nanjing based on a Baidu heat map, which can reflect the real-time population distribution, and then, we explored the spatiotemporal characteristics and spatial structure of the underground space use in urban built-up areas.

At present, in land use identification field surveys taking Shenzhen, China, as a case study, most complex land use identifications of categories A and B have been accurately matched to small categories, and the accuracy rate exceeds 85%. The general R, S, and U land use categories have been accurately matched to the medium category, and the accuracy rate exceeds 94%.

In this paper, we first use the business-weighted database for machine learning training through supervised classification learning. We have innovatively established a form database to store input items. During the parameter optimization process, based on the performance of the validation set, gradually adjust parameters such as the number of nodes, maximum depth, and learning rate to achieve optimal performance of the validation set under the same conditions. Then, we use the k-fold cross-validation method to avoid overfitting. The model clusters trained in machine learning training correspond to different city sizes. By setting parameters to determine the land area size of the target city, the corresponding classification model can be called. Finally, the feasibility of the proposed method is verified through experiments.

## 2. Urban Building Identification Method

### 2.1. Traditional Methods and Existing Problems

Urban land forms the basis of planning. The layouts of urban land have become progressively complex, increasing the technical difficulty and time costs of identifying the nature of urban land uses. However, the layouts and scales of urban land plots in cities of different scales and grades show evident differences; moreover, the differences in the business characteristics and architectural forms within the same type of land are increasingly evident. Identifying the type of urban land is the basis of various urban planning and design projects. By identifying urban land and analyzing the current construction characteristics, spatial forms, and land use structures of the city, we can conduct a physical examination, evaluations, planning, and design work.

The traditional method of urban land identification is to manually mark the land according to a comprehensive judgment of the buildings, business types, styles, public spaces, and so on of each plot. This can be performed by combining a current topographic map with a manual site survey by car, cycling, or walking, and then dividing the land and individually marking the land types in a vector database. Such recognition methods require a long time and large amounts of manpower and material resources. In addition, they rely on the human brain to judge whether the recognition standard is not uniform, and mixed or complex plots can easily lead to mistakes or arbitrary choices. Therefore, we should adopt automatic identification as the main method and manual verification as an auxiliary method for large-scale land use type identification at the urban scale.

### 2.2. Progress in Automatic Identification Methods

Currently, the most common automatic identification methods for urban land are POI big data and algorithm processing. The first method is a weight determination method that determines the type of plot by calculating the impact weight of each type of business. For example, Rong et al. [6] calculated the weights of various business types in a plot by calculating the density of various POIs in the plot and performing normalization processing. Then, they determined the land use nature of the plot based on weight priority. Anning calculated weights based on the frequency of various POIs in a plot and used a decision tree model to identify land uses in combination with remote sensing images. The second method involves entropy weighting. For example, Zhao et al. [8] used entropy weighting to calculate the functional intensity of various construction lands and then used the mean square deviation method to comprehensively judge their nature. The third method involves a mathematical model. For example, Zeng [10] used finite elements for the entire research area, arbitrarily selected a research unit, and established a linear discriminant analysis thematic mathematical model for the identification of land use types based on a certain attribute of a business-type point located in the unit.

Relative to mobile phone signaling, remote sensing images, bus-swiping data, and other data sources with higher access thresholds, business-type POI data can be directly obtained through network open-source data or purchased at a low cost. The data can be obtained immediately, effectively reducing research and application costs, and can be widely used in planning and preparation. Moreover, compared to other data, business-type data and their internal attributes are directly related to urban land types. To a certain extent, the credibility and accuracy of the recognition results can be guaranteed. However, identifying urban land using only business-type POI data ignores the correlations between urban building forms and urban land. In addition, the single data dimension and inconsistencies between the business-type data and national standard classifications can still lead to considerable errors in the final identification results. Moreover, this approach cannot achieve practical accuracy in identifying small urban areas. Based on this, this study combines business-type POI data with building form data and applies deep learning and fine recognition from the business-type data classification to the national standard land subcategory using AI technology.

### 2.3. Theoretical Breakthrough of Comprehensive POI and Architectural Form Identification of Urban Land

Different types of land use usually have different architectural layout patterns, as reflected in various form indicators and the layout of the three-dimensional space. Zhang et al. [16] discussed the growth mechanism of the block form from the perspectives of the height and plot ratio and found that building form indicators such as the maximum height of the building, average height of the building, average base area of the building, and plot ratio of the block change with changes in the land use function of the block. At the level of the various form indicators, Long et al. [17] divided a block into nine types based on analyzing the type of the block forms: “low density, low density, low density, low density, low density, medium density, high density, low density, high density, and high density”. Herein, it is necessary to establish a database of various building form index data and to discover the combined relationships of various form indices in different types of land through machine learning to improve the recognition accuracy.

The three-dimensional form layout of buildings can also reflect the land use type of a block. Martin et al. [18] summarized the geometric form characteristics of modern urban blocks with European cities as a reference and used them to obtain three-dimensional form models of different types of blocks. Winnie et al. [19] discussed the impacts of urban land use functions on the three-dimensional layouts of buildings in a block in FARMAX. When performing supervised classification, learning, and recognition, it is necessary to embed various 3D shape-intelligent rules. For plots with large buildings, mixed commercial and residential formats, and mixed office and commercial formats (i.e., those difficult to accurately identify using only business formats), the recognition results are intelligently and interactively judged through targeted 3D shape rules to further improve the recognition accuracy.

### 2.4. Advantages and Theoretical Basis of Urban Building Identification Method Based on Deep Learning

As the classification characteristics of POI data do not correspond to the land type (regardless of the algorithm used), a POI classification cannot be transformed into a land type. Therefore, it is necessary to add AI technology to supervised machine learning to classify and learn massive amounts of data. According to the number of business types in a plot, the combination method, and whether there is a strong correlation between the business type and various indicators of the form, the measured land use type results can be compared. Then, the internal relationships between the business form, building form, and land use type can be deeply studied to provide a finer identification of land use classifications.

Holzinger et al. [20] provided assistance for the success of AI in the fields of agriculture and forestry and identified three important frontier research areas. These included intelligent information fusion, robotics, concrete intelligence, and the extension, interpretation, and validation of trusted decision support. Holzinger et al. [21] also argued for using graph neural networks as a method-of-choice, enabling information fusion for multi-modal causality.

Supervised classification learning based on AI technology not only inputs the training data to the intelligent system but also delivers the correct classification results (labels of data) to the intelligent system for learning and analysis. Then, new and unrecognized data is released. In this way, the intelligent system can also calculate the probabilities of various results from the data and provide the most accurate result. The business-type POI data contains industry-type information and cannot be consistent with the classification results of the national land classification standard (GB50137-2011). Therefore, a supervised classification learning method must be adopted for the automatic recognition of urban land.

## 3. Workflow of AI to Identify the Appearance of Urban Buildings

### 3.1. Data Processing and Database Construction

(1)Multi-source big data collection. The business-type POI data uses network open-source data as the main data source. An industry classification of each point can be obtained through ArcGIS, including primary and secondary industry classifications. The primary classification includes 20 major categories, including “corporate enterprises, shopping, food, real estate, life services, medical care, government agencies, and finance.” The secondary classification is divided into several sub-categories based on the primary classification. Through research attempts and accuracy evaluations, this study chose to adopt a business-type database construction with first-level classification items. In addition, each business-type point also contains information such as business-type geographic coordinates, business-type projection coordinates, and business-type point names. The acquisition methods for architectural space vector data are relatively diverse and can be obtained through government departments or data providers in other relevant fields or directly downloaded and obtained through network open-source data. The building space vector data includes urban roads, urban blocks, and building vector data (including height information).(2)Calibration of business types and building spatial data. In general, the GPS positioning of POI vector data obtained from open-source data is accurate to six decimal places; thus, the data coordinates will have several meters of error and dislocation relative to the real space of the city. Therefore, in the case of unified projection coordinates between business-type data and building spatial data, an inevitable slight drift remains in the spatial coordinates, as shown in Figure 2. In this study, expansion and spatial connection methods are used to perform an automatic spatial correlation between the plots and business sites to correct the deviation.(3)Weighted transformation of business data. This study needs to conduct an in-depth analysis of the business-type data and land use types of the case cities. This requires that the number of various business-type sites be in the same or an adjacent order of magnitude. However, the numbers of various industries in a real urban space are very different and cannot be maintained at the same level. Table 1 lists the information classifications of business types in Shenzhen as an example. The numbers of “corporate enterprises”, “shopping”, and other business types in the main urban area of Shenzhen exceed 100,000, whereas the numbers of “scenic spots”, “cultural media”, and other business types reach only near 5000. The massive difference in the order of magnitude can easily cause business-type points with small orders of magnitude to be obliterated by other data in the deep learning and calculations. Additionally, differences exist in the importance weights of different types of business site data. For example, for a financial and insurance (B21) plot, the number of commercial retail stores belonging to B11 is far greater than the number of financial stores. This shows that the importance of the financial business-type points for this plot is greater than that for commercial retail stores.

Based on this, the term frequency (TF) method is used to normalize the characteristic data of a business form. This is based on the ratio of the characteristic frequency of a business form in a plot to the characteristic frequency of that business form in the whole city. The inverse document frequency (IDF) method is used to measure the importance of the business characteristics and is obtained from the logarithmic value of the ratio of the total number of plots in the city to the number of plots containing these business characteristics. The TF-IDF algorithm is as follows:ni,j∑Kni,j⋅logDj:ti∈Dj
where i is the plot number, j is the format feature number, ni,j is the frequency of format feature j of plot i, K is the dimension of the format feature, D is the total number of plots in the city, and j:ti∈Dj is the number of plots whose frequency of format feature j is not 0.

(4)Database construction. The characteristics of each business type are reweighted according to the frequency and importance of their occurrence. Their weighted values are entered into the business-type attribute table, and the business-type POI database is constructed. The production details include the GPS positioning coordinate data of all business-type points, industry classification attributes of all business-type points, the name of each business-type point, and the TF-IDF weight value of each business-type point. As aforementioned, several existing urban land use recognition methods based on big data ignore the correlations between the urban architectural form and urban land use. As a result, evident deviations occur in the identification of parts of the land. Moreover, such land use identifications are not accurate for the middle class or even the small class. Based on this, this study calculates four morphological indicators: the maximum building height, average building height, average building base area, and plot ratio for each plot. The weighted business-type data are spatially associated on the plot to build a database containing the weighted business-type features, which are architectural form features, and are incorporated into the training module of the machine learning to improve the accuracy of the recognition.

### 3.2. Automatic Recognition of Urban Buildings Based on Supervised Classification Learning

Algorithm performance comparison and filtering. Common supervised learning classification algorithms include logical regression models, tree models, support vector machines, and integration models. Considering that the parcel labels are divided by sub-categories, the number of categories is large, and the characteristic dimensions of the parcel are high (including not only the weighted business-type characteristic dimension but also the architectural form characteristic dimension of the parcel). To improve the classification accuracy of the model, this study compared the classification accuracy of the support vector classifier, Lasso model, gradient-boosted decision tree (GBDT), and other models. The performance comparison is shown in Figure 3. After comparing the results, this study chose the optimized GBDT gradient-lifting decision tree model for classification. In general, a gradient-boosting algorithm is a machine learning technology used for regression, classification, and sorting tasks and belongs to the boosting algorithm family. A gradient-boosting algorithm builds a learner at each step of the iteration to reduce losses along the steepest direction of the gradient to compensate for the shortcomings of existing models. The classic AdaBoost algorithm can only handle binary learning tasks using exponential loss functions, whereas a gradient-lifting method can handle various learning tasks (multiclassification, regression, ranking, etc.) by setting different differentiable loss functions, thereby greatly expanding the application range. The gradient-lifting algorithm uses the negative gradient of the loss function as a residual fitting method. If the basis function in the algorithm uses a decision tree, a GBDT can be obtained. In the process of parameter optimization, the number of nodes, maximum depth, learning rate, and other parameters are gradually adjusted according to the performance of the verification set, such that the performance on the verification set is optimum under the same conditions (Figure 4). Simultaneously, the k-fold cross-validation method is used to avoid overfitting the model. The original dataset is divided into training and test sets to avoid overfitting on the training set in the pursuit of high accuracy, thereby increasing the prediction accuracy of the model on data outside the sample. However, differences in the division of training and test sets can cause significant changes in the accuracy of the model. To eliminate this variable factor, we create a series of training and test sets, calculate the accuracy of the model for each test set, and then calculate the average value. This is the essence of k-fold cross-validation.

Supervised classification and in-depth learning. This study selects a city with the same scale as the target city as the learning sample, obtains the business-type weighted database, and builds a form database of the sample city through the above multi-source data collection, spatial calibration, and weighted transformation methods. Simultaneously, the land use data of the sample city (divided by the standard subcategory of the nature of the land use) is used as the machine learning label, and the GBDT gradient-elevation decision tree is used for the supervised classification learning. We require: Learning dataset {xi,yi}i=1n, number of iterations: m, Loss function Ly,Fx; learning rate v. Ensure: The predict model: Fx.
F0x=0for t=1→mfor i=1→ndo y′i=−∂Lyi,Fxi∂FxiFx=Ft−1x

Build a new decision tree hix according to {xi,y′i}i=1n
ρt=argminρ∑i=1nLyi,Ft−1x+ρ∗htxFtx=Ft−1x+v∗ρt∗htxend forOutput the final model Fx=∑i=1mFix

The business-type weighted database and building form database are used as the data input items for the machine learning training. Correlations are identified between the type of business site, the name of the business site, various form indicators, other data about each plot, and the actual land use type of the plot. This information is used to generate the machine-learning classification model. The number of training samples in the classification model affects the accuracy of the final recognition results to a certain extent. According to our team’s experience in the experiment, when the number of training samples exceeds 5000 plots, the accuracy of the recognition results often exceeds 80%. Sample cities of different scales and grades can be combined to form urban land automatic recognition model clusters. Independent models can be trained according to the city size of the city to meet the needs of various types of city recognition.

Automatically identify the land type. Finally, the AI system conducts automatic recognition and calls the model clusters corresponding to the various city sizes as trained in machine learning. By setting the parameters to determine the size of the area that the target city needs to identify, the corresponding classification model can be automatically called. When the system needs to automatically identify the land use of a target urban area, the corresponding classification model is selected according to the urban size of the target urban area. By calling the database, the collected commercial characteristics and architectural form characteristics of the target urban plot are obtained and input into the classification model as feature vectors. Finally, the possible land use categories for each plot in the target urban area are generated. After obtaining the corresponding land use properties of each plot, the plots of the same type of land use properties are filled with one color (with the precision of fine to small land use properties). In addition, the system automatically marks the confidence of each land use recognition result. The nature of the marked land can adopt the urban construction land classification of The Standard for Classification and Planning of Urban Construction Land; the urban classification land standard of each region and city can also be used.

### 3.3. Human-Computer Intelligent Interaction and Feedback Optimization

#### 3.3.1. Multi-Type City Sample Learning

Using a large number of sample city databases and lands as learning labels to identify the land type of a target city can effectively improve the accuracy of the urban land intelligent recognition system. In this study, the selection of urban samples is further divided according to the sizes and locations of cities to select reasonable sample cities for different target cities. According to The Notice of the State Council on Adjusting the Criteria for the Classification of City Size, sample cities can be divided into five categories: small cities, medium-sized cities, large cities, and two types of megacities. For each type of city, several typical cities are selected as examples for the machine learning. Then, the sample cities of each category are further divided into sub-categories such as mountain cities, plain cities, and coastal cities according to their location to achieve accurate matching of the learning samples. This can effectively reduce the cost of data acquisition and learning while improving the recognition accuracy.

#### 3.3.2. Intelligent Rules

In addition to the weighted processing of the business-type POI data, superposition of the building form database, and supervised machine learning, specific intelligent rules are also added. This can effectively improve the precision and accuracy of the land use recognition.

The first rule is an indicator rule. Specifically, when the shape indicators, such as the floor area and height of a certain plot, are within a specific numerical range, the intelligence is mapped to a specific land type. For example, the business type of “corporate enterprise” includes industrial and commercial types. “XXX Textile Co., Ltd.” and “XXX Automobile Processing Company” belong to the industrial land category. “XXX Consulting Co., Ltd.” and “XXX Business Co., Ltd.” belong to the commercial land category. Therefore, after machine learning and recognition, mistakes can easily be made in the recognition of Class M and Class B2 lands. In view of this phenomenon, the method for distinguishing the two types of land use is based on the fact that there are significant differences in the building height and bottom area of the land where they are located. Accordingly, an indicator rule is added as a limiting condition. For example, if there is a building block with a floor area of not less than 1500 m^2^ and a building height of not more than 12 m on a certain plot containing a business type of “company enterprise,” the plot is classified as industrial M or logistics W land. Otherwise, it is B2 land.

The second rule concerns the architectural layout form. For residential plots, industrial park plots, industrial office building plots, and other plots with evident layouts (Figure 5) and commercial plots with enclosed layouts, a fuzzy judgment is made by introducing shape indicators such as the building spacing, dissimilarity index [22], enclosure degree [23], and building orientation [24]. Then the form label of the plot is given to calibrate and improve the accuracy of the business-type identification.

The third rule concerns special architectural forms. For buildings with special shapes, such as multiple plot combinations, the bottom area, height, and height difference of each block are calculated and identified using clustering. Then, a specific architectural form label is provided. For example, complex buildings are usually a combination of low- and high-rise blocks. The system calculates the height difference and bottom area of the podium and tower. When the data meets a specific interval, the system can intelligently identify whether this type of building is a complex building. Then, it can be judged that the land type belongs to B2; it can be further identified into sub-categories according to the type of business.

#### 3.3.3. Interactive Feedback and Self-Optimization

After completing the database construction, supervised classification learning of multi-level sample cities, and automatic recognition, and intelligent rule calibration, the AI system can automatically identify the land type. The corresponding confidence degree of each type of land is given via comparison with a real and practical land database. A field survey is conducted in the target city (focusing on the plots with low confidence) to check the identification results one-by-one and feedback the wrong plot type information to the system. The feedback results are combined with business data and form data for further clustering learning to improve the accuracy of the next round of recognition. Through the identification and interactive feedback for different cities, the sample database can be continuously enriched to realize the system’s self-optimization feedback and correction.

## 4. Experiment

### 4.1. Case Selection

Shenzhen is a super-city, with a population of over 10 million. Knowledge-based service industries, such as those concerning high-tech, financial securities, and trade, are developing rapidly. Many urban areas represented by Futian central business district have seen a large number of industrial clusters, highly complex and mixed land uses, and frequent changes in land use types. Therefore, Shenzhen is taken as the research sample. The AI system is constructed by integrating the business-type POI and building form big data to accurately identify the attribute types of Shenzhen’s urban lands. An annual basic database of Shenzhen’s urban land is built to reflect the rapid changes in industries; it also provides experience for planning and data updates for other cities.

Based on the business-type POI and building space big data of Shenzhen in 2018 (Figure 6), an expansion space calibration was initially conducted for the business-type point layer and plot-unit layer (Figure 7). The numbers of various types of business points owned by the space unit of each plot (i.e., the frequency of business characteristics of each plot) are calculated. A parcel attribute table of the associated business-type point is generated. Then the TF-IDF algorithm is applied to the completed plot database of the associated business-type points, and the characteristics of each business type are reweighted according to their occurrence times and importance. Based on this, a parcel feature attribute library containing weighted business features is obtained. Finally, the building layer in the building space data is spatially connected with the parcel unit layer. The multiple building form indicators of each parcel are calculated and added to the parcel feature attribute library containing the weighted business characteristics. Then, the system generates a “sample city database.” In this study, the research team also cooperated with the Shenzhen Urban Planning and Design Institute Co., Ltd. (Shenzhen, China) Based on vector data, accurate data on the sample land was obtained through a survey. The land use data obtained through in-depth learning and recognition was compared with the accurate data on a point-by-point basis. The data results were analyzed, and further optimization was conducted through intelligent interaction rules.

### 4.2. Deep Learning Training Process

The supervised classification learning algorithm is used to conduct machine learning training on the sample city database (including the weighted business and architectural form characteristics of each sample city). The characteristics are combined to form an urban land automatic recognition model cluster. The “target city database” of Shenzhen is entered into the system, and the system identifies and outputs the land use nature of each plot in Shenzhen (Figure 8).

Based on the perspective of trusted AI, the interpretability of identifying land types from the two perspectives of the business-type POI and building form can be further explored. According to the interpretability algorithm proposed by Friedman [25], the correlation between the land use identification results and business-type POI and building form can be respectively analyzed. Among them, the abscissa represents the proportion of the number of variables in the input test set (such as the number of business types in government agencies) to the total number of variables. The ordinate indicates the correlation between the land type and the variable (e.g., the correlation between A1 administrative office land and government agencies or that between the R2 Class II residential land and plot ratio).

It can be seen from Figure 9 that clear correlations exist between the land use and business types. For example, there is a strong correlation between A1-type land and government agency-type business data, B29-type land and company enterprise-type business data, and B32-type land and sports and fitness-type business data. Notably, the other parts of the land use and business types do not have one-to-one correspondences, but the combination of multiple business types in terms of quantity and frequency is related to the land use.

Figure 10 illustrates that there is a certain relationship between land use types and architectural forms. Although the overall correlation is not as strong as that of business types, strong or weak correlations exist with the building height, density, and other indicators. This also demonstrates the feasibility of integrating business and building form data. In some cases, the relationship between the land use and building form is very weak, such as when land is used for social welfare facilities. Owing to the different levels and scales of welfare homes, orphanages, and nursing homes, the base areas, plot ratios, and heights of some community-level social welfare buildings are relatively small. Accordingly, significant differences exist in the form of indicators between large apartment nursing homes and other buildings. Therefore, it is difficult to optimize and improve the accuracy of land use identification in these cases by using building forms.

### 4.3. Intelligent Interaction Rules and Optimization

After a field survey of the Nanshan, Futian, and Luohu Districts in Shenzhen, the recognition accuracy rate of each type of land was calculated by comparing the identified results (Table 2). As shown, the recognition of Class A and Class B land is accurate for small categories, and the overall accuracy rate exceeds 80%. The R, M, W, S, and U categories of land are accurate to the middle category, and the overall accuracy rate exceeds 85%. To further improve the accuracy of the land use recognition for the R, A, B, and M categories, the intelligent interaction rules for the indicators and forms are integrated. Then, further algorithm optimization is conducted for R2, B2, A3, M, and other types of land.

Many R2 plots are identified as B1 plots. Through a field survey, it was found that most of these residential areas are villages in the city. These are usually densely arranged, without sunlight spacing, and have a large number of shops arranged along the street (Figure 11, left). As a result, their business characteristics, maximum building height, average building height, and other indicators are quite similar to those of Class B1 plots. Moreover, the difference between the block and a B1 block is characterized by the high degree of block enclosure and the layout of the business types along the boundary of the block (Figure 11, right). Accordingly, a rule concerning the building layout and form can be added to the system: “If the identified B1 plots (i.e., B11 to B14 plots) have a degree of enclosure ≥ 70%, the number of buildings exceeds 15 buildings/hm^2^, and more than 70% of the business types are concentrated within 10 m along the street, the identification result is converted to R2 land”.

Similarly, owing to the influences of catering and other business forms, many A33 schools in Shenzhen are mistakenly identified as Class B1 land; this is because these schools have highly differentiated playgrounds (Figure 12). Based on this feature, special morphological rules are added to the recognition system. The neural network data dimension reduction processing method proposed by Jeffrey et al. [26] is used to extract and cluster the autoencoder feature of the linear data of the playground shape in the database to provide intelligent recognition of the playground shape. When the form of a playground appears in the plot, the plot is directly determined to be a non-B1 plot. On this basis, it is further divided and identified according to the names of the business types.

Finally, after machine recognition and the multi-round interactive optimization feedback of the intelligent rules, the final land use recognition result for Shenzhen is obtained (Figure 13). This research method is then extended to the identification of urban land of different types and sizes in Nantong, Changzhou, Weihai, and other cities (Figure 14). While further enriching the sample database, new intelligent rules are constantly incorporated to further improve the recognition accuracy.

Currently, in the recognition of land, the recognition of Class A and Class B land is accurate for small categories, and the overall accuracy rate exceeds 85%. Classes R, S, and U land are accurate for the middle class; the overall accuracy rate reaches more than 94%. The above results indicate that the proposed method outperforms previous methods in the recognition accuracy of Classes R, S, and U land. In addition, for cities where land detection and land use information are relatively complete in planning and urban management, such as Shanghai, Shenzhen, and Nanjing, the land use identification method proposed in this study can be applied to quickly obtain the latest urban land use function data. Then, the construction of the urban land can be monitored and scientifically evaluated in real time. By comparing the differences between the planned land use type and the actual construction situation of the plot, the impact mechanisms of the project can be studied and an optimization strategy can be proposed. This helps urban construction more closely approach the requirements of various business types, quantities, scales, and other indicators in the planning process so as to improve the land’s environmental benefits.

## 5. Conclusions

This study proposes an automatic recognition method for multidimensional urban building appearances based on the distribution characteristics of urban building data and 3D entities. Compared with the traditional method of field surveys to identify land use, this study can effectively avoid the problems of long survey times, large investments of manpower and material resources, and easy misjudgments regarding the construction of complex plots.

Compared with the existing methods of identifying urban land through POI big data, the breakthrough point of this study is that through supervised machine learning and comprehensive building form data, the building types can be accurately identified for medium or small categories, rather than being limited to residential, commercial, office, and other urban land categories. The recognition accuracy and precision are further improved.

Nevertheless, owing to limitations concerning the business type, building form data, and team knowledge level, the following problems need to be addressed in the future: First, because the residential environment and building quality cannot be directly reflected by the business type and building form information, it is impossible to completely distinguish R-type land only based on the building density, height, number of public service facilities, and other information. In particular, low-rise commercial housing and temporary housing facilities are similar in shape. Simultaneously, the number of public and service facilities in the residential area is relatively small; nevertheless, certain facilities are located at the bottom of the building. This increases the difficulty of the fine recognition of Class R land, especially for small classes such as R22. In view of this problem, we believe that we can incorporate architectural textures in the next step of research to distinguish between land types with similar architectural forms and large layout differences (e.g., low-rise commercial housing and temporary housing). This can be accomplished using the texture differences between homogenized and irregular layouts. With the continuous progress of digital technology in the future, big data will become increasingly multi-sourced and trusted AI will receive increased attention. As such, the academic community should explore how to use multi-source big data combined with AI technology in urban research and how to apply it deeply to the actual work of planning.

## Figures and Tables

**Figure 1 sensors-23-05353-f001:**
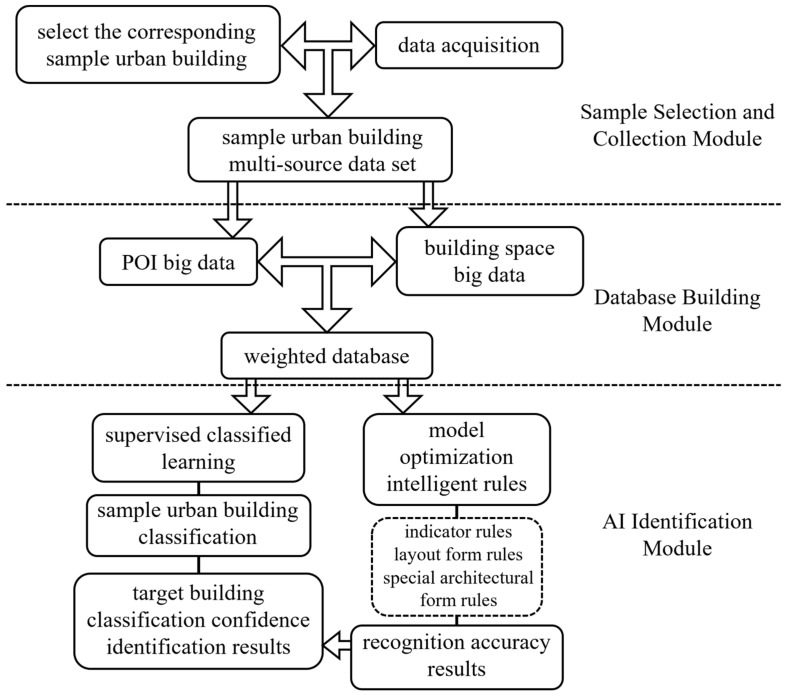
Technical framework for precision identification of urban land use type using artificial intelligence.

**Figure 2 sensors-23-05353-f002:**
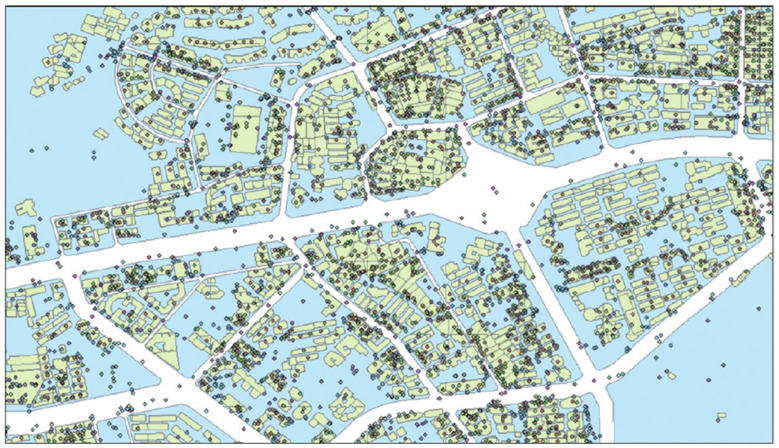
Spatial deviation between business and architectural space data.

**Figure 3 sensors-23-05353-f003:**
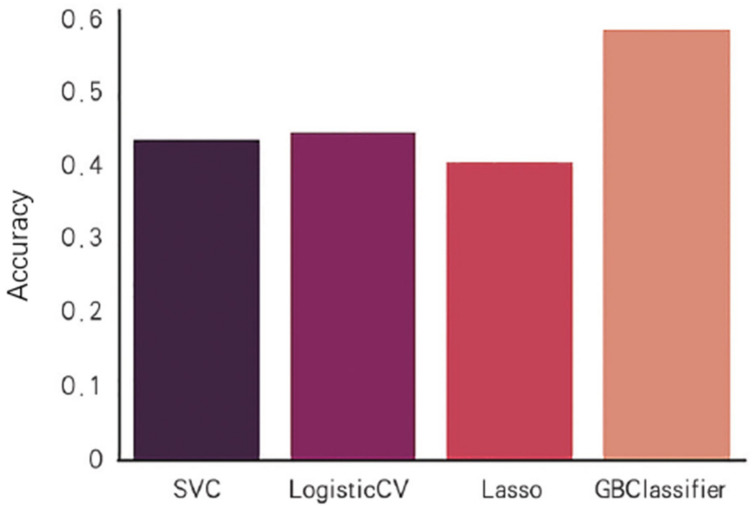
Performance of each algorithm.

**Figure 4 sensors-23-05353-f004:**
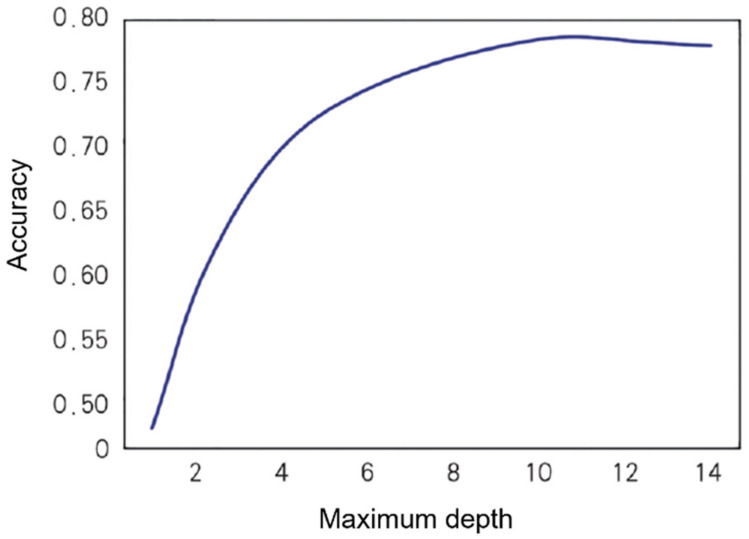
Effect of optimized gradient-boosted decision tree (GBDT) algorithm.

**Figure 5 sensors-23-05353-f005:**
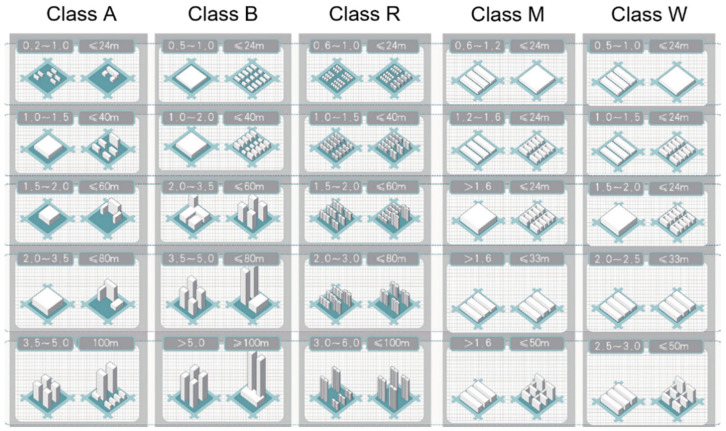
Architectural layouts of various types of land uses (Values in the figure represent the plot ratio and height, respectively).

**Figure 6 sensors-23-05353-f006:**
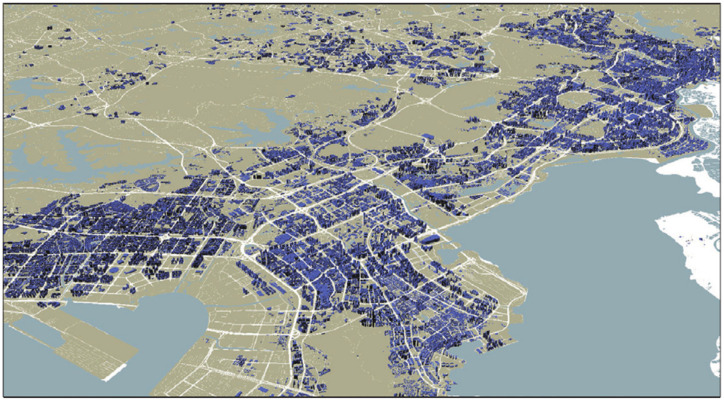
Bird’s-eye view of big data of architectural spaces in Shenzhen.

**Figure 7 sensors-23-05353-f007:**
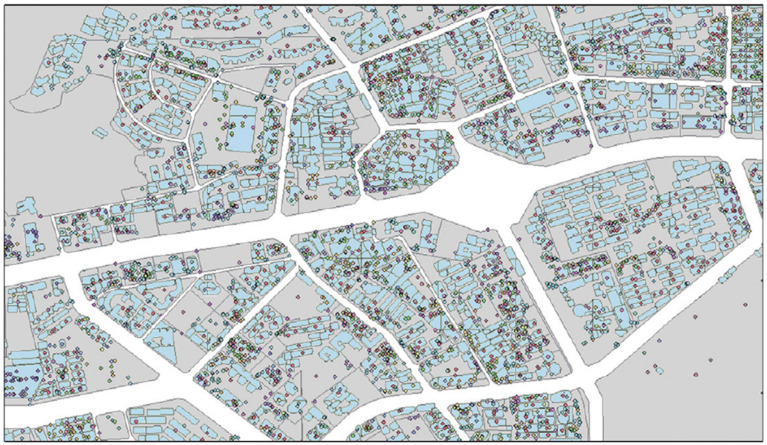
Spatial calibration of points of interest and architectural space data in Shenzhen.

**Figure 8 sensors-23-05353-f008:**
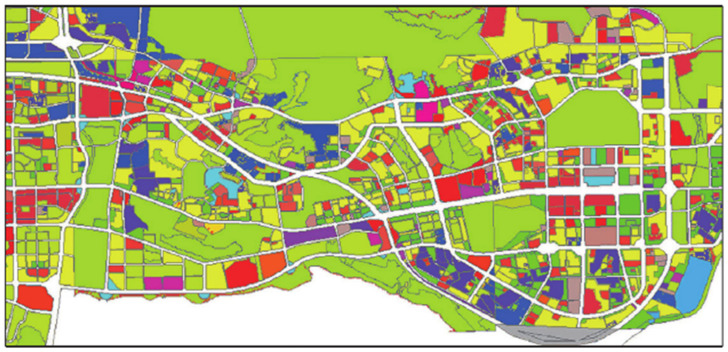
Land use identification results of Shenzhen after learning and training of sample cities (*part*).

**Figure 9 sensors-23-05353-f009:**
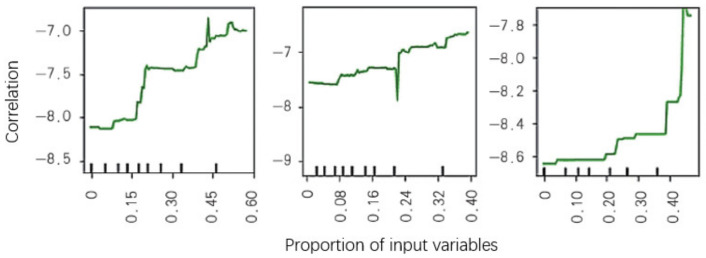
Correlations between land use type and industry classification (The (**left**) figure shows A1 land and government agencies, the (**middle**) figure shows B29 land and corporate businesses, and the (**right**) figure shows B32 land and sports and fitness businesses).

**Figure 10 sensors-23-05353-f010:**
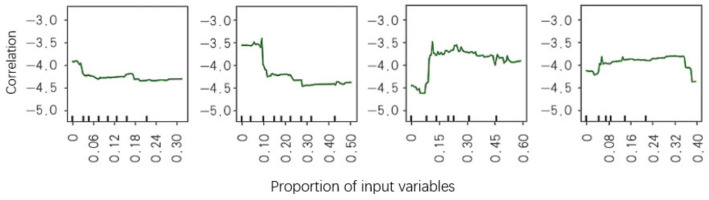
Correlations between land use type and building form index (The (**leftmost**) figure shows the maximum height of R2 land and buildings, the (**second-from-left**) figure shows the average height of R2 land and buildings, the (**second-from-right**) figure shows the average base area of R2 land and buildings, and the (**rightmost**) figure shows the plot ratio of R2 land and plots).

**Figure 11 sensors-23-05353-f011:**
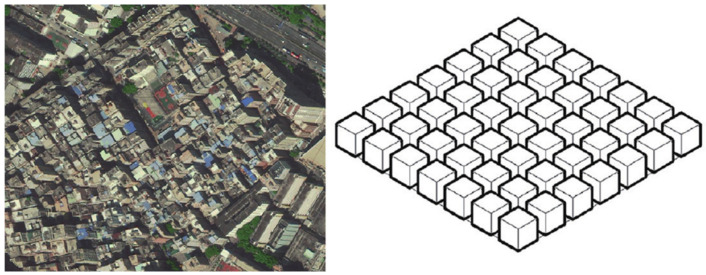
Compact residential layout in Shenzhen (The left figure is the image of the residential area derived from Baidu Maps, and the right figure is the abstract pattern of the form layout).

**Figure 12 sensors-23-05353-f012:**
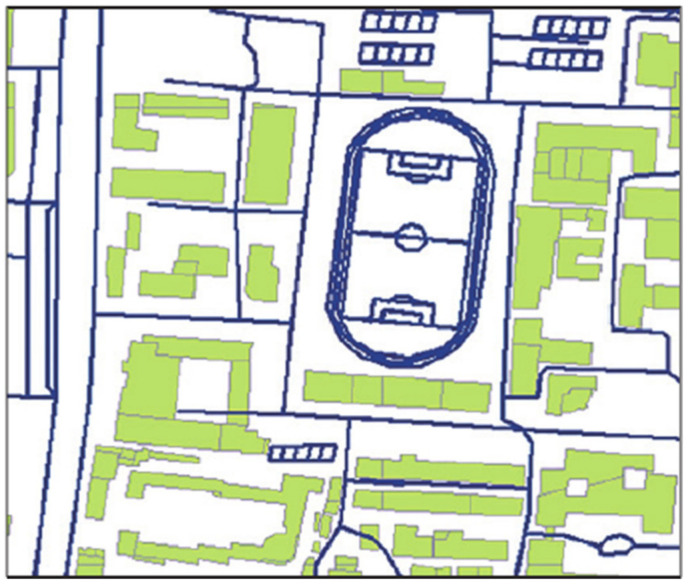
Characteristics of the special shape of the school.

**Figure 13 sensors-23-05353-f013:**
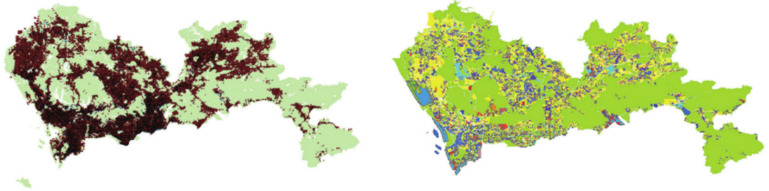
Original data of business types and architectural space in Shenzhen (**left**) and intelligent identification results for land uses in Shenzhen (**right**).

**Figure 14 sensors-23-05353-f014:**
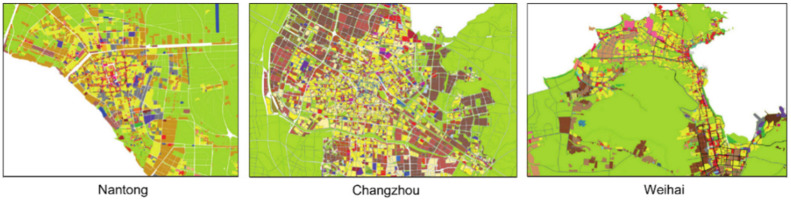
Identification results for land uses in sample cities of different sizes in China.

**Table 1 sensors-23-05353-t001:** Numbers of various types of businesses in Shenzhen.

Building Type	Company	Shopping Mall	Fine Food	Doorway	Real Estate
Quantity	131,026	104,681	90,860	79,259	77,806
Building type	Life service	Road	Transportation facilities	Beauty salon	Government
Quantity	64,810	35,580	32,851	29,738	25,629
Building type	Automobile service	Medical center	Finance and banking	Places of entertainment	Hotel
Quantity	23,863	19,853	19,806	18,372	14,187
Building type	Exercise and fitness	Education	Scenic spot	Cultural media	Natural features
Quantity	7518	6478	5842	5643	317

**Table 2 sensors-23-05353-t002:** Accuracy rates of construction land type identification in Shenzhen.

Building Type	R1	R2	R3	A1	A21	A22	A31	A32	A33	A34	A35	A41
Accuracy	0.861	0.853	0.859	0.856	0.901	0.938	0.88	0.856	0.818	0.82	0.859	0.926
Building type	A42	A51	A52	A53	A59	A6	A7	A8	A9	B11	B12	B13
Accuracy	0.921	0.818	0.841	0.89	0.879	0.834	0.91	0.927	0.903	0.798	0.796	0.802
Building type	B14	B21	B22	B29	B31	B32	B41	B49	B9	M1	M2	M3
Accuracy	0.825	0.839	0.823	0.799	0.813	0.89	0.916	0.851	0.833	0.823	0.891	0.831
Building type	W1	W2	W3	S1	S2	S3	S4	S9	U1	U2	U3	G
Accuracy	0.813	0.781	0.797	0.989	0.935	0.951	0.907	0.891	0.862	0.848	0.865	0.746

(Class A and B land are identified as accurate for the small class, R, M, W, S, and U lands are identified as accurate for the medium class, and G land is identified as accurate for the large class).

## Data Availability

The data used to support the findings of this study are available from the corresponding author upon request.

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
