# Peer review of "Classification and Recognition of Building Appearance Based on Optimized Gradient-Boosted Decision Tree Algorithm"

_sensors, 2023, doi:10.3390/s23115353_

Round 1
Reviewer 1 Report
Summary:
This paper proposes an optimized gradient-boosted decision tree algorithm to enhance a decision tree model for building classification. Through supervised classification learning; machine learning training was conducted using a business-type weighted database. A form database was built to store the input items. During parameter optimization; parameters such as the number of nodes; maximum depth; and learning rate were gradually adjusted based on the performance of the verification set to achieve optimal performance on the verification set under the same conditions.
Strengths:
-The proposed model is well-detailed, which makes it easier to replicated the reported results
-Throughout expirments, which validates the proposed method and support the conclusions
Weaknesses:
-contributions are not well-highlighted
-some related works are missing
Comments:
-add paper contributions in form of bullets list in the introduction
--at the end of introdution section, add 'paper outline' paragraph that describe the content of each section
-Although Figure 3 shows the superiority of GBDT gradient-lifting decision tree model for classification in terms of Accuracy, it would be good if you add the recall figure as well, that is because some models perform very well in terms of accuracy, but have very low recall.
-Figure 5 text is not clear at all, it appear like it was take as a screenshot, change it to text instead of an image
-Some related works are missing:
[1] Rahman, Ashikur, et al. "Performance of different machine learning algorithms on satellite image classification in rural and urban setup." Remote Sensing Applications: Society and Environment 20 (2020): 100410.
[2]Chaib, Souleyman, et al. "On the Co-Selection of Vision Transformer Features and Images for Very High-Resolution Image Scene Classification." Remote Sensing 14.22 (2022): 5817.
Author Response
Reply to reviewer 1
- The content of the introduction section need significant revision.
REPLY:
Thank you for your suggestion. The introduction section has been rewritten.
- At the end of introdution section, add 'paper outline' paragraph that describe the content of each section.
REPLY:
Thank you for your suggestion. We have added "Thesis Outline" at the end of the introduction section to make the paper more clear. The details are as follows.
In this paper, we first use the business weighted database for machine learning training through supervised classification learning. We have innovatively established a form database to store input items. During the parameter optimization process, based on the performance of the validation set, gradually adjust parameters such as the number of nodes, maximum depth, and learning rate to achieve optimal performance of the validation set under the same conditions. Then, we use the k-fold cross validation method to avoid overfitting. The model clusters trained in machine learning training correspond to different city sizes. By setting parameters to determine the land area size of the target city, the corresponding classification model can be called. Finally, the fea-sibility of the proposed method is verified through experiments.
- Although Figure 3 shows the superiority of GBDT gradient-lifting decision tree model for classification in terms of Accuracy, it would be good if you add the recall figure as well, that is because some models perform very well in terms of accuracy, but have very low recall.
REPLY:
Thank your for your suggestion. The model mentioned in this paper excels in terms of "classification accuracy", but lacks recall compared to other methods. Nevertheless, the method proposed in this paper also has certain research value, and in future research, some models that can improve recall rates will be considered.
- Figure 5 text is not clear at all, it appear like it was take as a screenshot, change it to text instead of an image.
REPLY:
Thank you for your suggestion. We have changed the screenshot of this section to text.
- Some related works are missing:
[1] Rahman, Ashikur, et al. "Performance of different machine learning algorithms on satellite image classification in rural and urban setup." Remote Sensing Applications: Society and Environment 20 (2020): 100410.
[2]Chaib, Souleyman, et al. "On the Co-Selection of Vision Transformer Features and Images for Very High-Resolution Image Scene Classification." Remote Sensing 14.22 (2022): 5817.
REPLY:
Thank you for your suggestion. We have added the two references you provided to the paper. The details are as follows.
RAHMAN et al. main goal was to describe the performance of different machine learning algorithms on three different spatial and multispectral satellite image classification in rural and urban extents. And they used Random forest, Support Vector Machine (SVM), and their combined strength were applied on Landsat-8, Sentinel-2, and Planet images separately to assess individual and overall class accuracy of the images [22]. CHAIB et al. proposed a new deep framework is proposed for very high-resolution(VHR) scene un-derstanding by exploring the strengths of Vision transformer(ViT) features in a simple and effective way. This pre-trained ViT models are used to extract informative features from the original VHR image scene, where the transformer–encoder layers are used to generate the feature descriptors of the input images [23].
Reviewer 2 Report
The detailed comments are attached.

I am of the opinion that it has the potential to significantly advance the field. Nevertheless, there are several major revisions that must be made prior to acceptance for publication.
Author Response
Reply to reviewer 2
- Some numerical outcomes of the study should be highlighted in the abstract.
REPLY:
Thank you for your suggestion. Some experimental data results have been added to the paper. The details are as follows.
The experimental results show that this algorithm has high accuracy in building recognition. Es-pecially in R, S, and U class buildings, the overall accuracy rate of recognition reaches over 94%.
- What was the specific aim of the optimized gradient-boosted decision tree algorithm in the study?
REPLY:
Thank you for your suggestion. The specific purpose of studying this method is for better application. The gradient enhanced decision tree algorithm was chosen because it can achieve high accuracy with relatively little parameter tuning time. It can flexibly handle various types of data, including continuous and discrete values, and has a wide range of applications, making it suitable for the research in the field of architecture in this paper.
- How effective was the k-fold cross-validation method in avoiding overfitting?
REPLY:
Thank you for your suggestion. This verification method is the relatively optimal model obtained through multiple comprehensive comparisons, which can avoid the problem of overfitting to a certain extent.
- Can you explain how the classification model was invoked by setting the parameters to determine the size of the area of land for a target city?
REPLY:
Thank you for your suggestion. The parameter setting of the system first selects the corresponding classification model based on the city size of the target urban area. By calling the database, the collected commercial and architectural features of the target urban plot are obtained, which are input as feature vectors into the classification model. Finally, generate specific parameters for each plot in the target urban area. These parameters include parcel category, dimensional features, original location data, etc. After a large amount of training to obtain the parameters corresponding to different types of land parcels, the same type of land parcels are filled with a color with precision ranging from fine to small class land use properties, and the confidence level of each land use recognition result is automatically annotated. The size of its land area generally depends on the confidence level during training.
- 5.There are some serious problems with the introduction section.
- a) Use journal style for references
- b) The comparison of the previously used techniques and the advancement by the presented technique must be clearly highlighted.
- c) The literature should be improved by acknowledging recent studied related to machine learning and neural networks for example: a single-stage 3D object detection network based on density clustering and graph neural network, Subsurface sedimentary structure identification using deep learning: A review. Earth-Science Reviews...
- d) These potential studies must be studied in the manuscript.
REPLY:
Thank you for your suggestion. The introduction section has been revised. Added potential research directions and added multiple references you provided to this paper. We have selected references related to this paper from the literature you recommended above, and some of them are cited as follows. Xiong et al. [24] proposed a novel single-stage 3D object detection network based on density clustering and graph neural networks. Which utilizes density clustering ball query to partition the point cloud space and exploits local and global relationships by graph neural networks. Zhan et al. [25] describes how DL-based methods have been utilized for subsurface sedimentary structure identification from the viewpoint of different identification approaches (direct and data assimilation-based modeling) and differences between DL-based and traditional methods are discussed.
In the related research of urban architecture and space. Chen et al. [26] analyzed the relationship between the population distribution and the underground space use of the central city of Nanjing based on a Baidu heat map, which can reflect the real-time pop-ulation distribution, and then, we explored the spatiotemporal characteristics and spatial structure of the underground space use in urban built-up areas.
Some existing problems and potential research directions are as follows.Nevertheless, owing to limitations concerning the business type, building form data, and team knowledge level, the following problems need to be addressed in the future. First, because the residential environment and building quality cannot be directly reflected by the business-type and building form information, it is impossible to com-pletely distinguish R-type land only based on the building density, height, number of public service facilities, and other information. In particular, low-rise commercial housing and temporary housing facilities are similar in shape. Simultaneously, the number of public and service facilities in the residential area is relatively small; never-theless, certain facilities are located at the bottom of the building. This increases the difficulty of the fine recognition of Class R land, especially for small classes such as R22. In view of this problem, we believe that we can incorporate architectural textures in the next step of research to distinguish between land types with similar architectural forms and large layout difference (e.g., low-rise commercial housing and temporary housing). This can be accomplished using the texture differences between homogenized and ir-regular layouts. With the continuous progress of digital technology in the future, big data will become increasingly multi-sourced and trusted AI will receive increased attention. As such, the academic community should explore how to use multi-source big data combined with AI technology in urban research and how to apply it deeply to the actual work of planning.
- The quality of some figures must be improved.
REPLY:
Thank you for your suggestion. We have improved the quality of some of the images.
- Some novel aspects of this study must be included at the end of the introduction section.
REPLY:
Thank you for your suggestion. The innovation of this paper has been written at the end of the introduction section. The details are as follows.
Compared with the existing methods of identifying urban land through POI big data, the breakthrough point of this study is that through supervised machine learning and comprehensive building form data, the building types can be accurately identified for medium or small categories, rather than being limited to residential, commercial, office, and other urban land categories. The recognition accuracy and precision are fur-ther improved.
Reviewer 3 Report
Please check the semicolon punctuation in the abstract.
The abstract is disqualified. The major achievement of this paper is not clarified. The terminology in the abstract is not consistent with the title for the phrase of deep neural network and machine learning.
The introduction is lengthy and not informative to clarify the novelty of this work, how the previous work inspired the authors.
The authors should highlight the area in figure 2 to demonstrate the features that the authors wanted the readers to see.
Please provide the legend for figure 4.
Please list the label of x axis in figure 10 and 11.
Please try to combine some figures with multiple sub panels to make the figures more informative. Please add some description in the figure caption so that reader can know what the authors wanted to show.
Please provide a schematic of deep learning structure.
Please consider the language editing service or seek the help of native speaker.
Author Response
Reply to reviewer 3
Please check the semicolon punctuation in the abstract.
REPLY:
Thank you for your suggestion. The "semicolon" in the paper has been changed to "comma".
The abstract is disqualified. The major achievement of this paper is not clarified. The terminology in the abstract is not consistent with the title for the phrase of deep neural network and machine learning.
REPLY:
Thank you for your suggestion. We have rewritten the abstract section and changed the previously inappropriate topic to “Classification and Recognition of Building Appearance Based on Optimized Gradient-Boosted Decision Tree Algorithm”.
The introduction is lengthy and not informative to clarify the novelty of this work, how the previous work inspired the authors.
REPLY:
Thank you for your suggestion. The introduction and conclusion sections have been rewritten. The highlight in the following content.
Compared with the existing methods of identifying urban land through POI big data, the breakthrough point of this study is that through supervised machine learning and comprehensive building form data, the building types can be accurately identified for medium or small categories, rather than being limited to residential, commercial, office, and other urban land categories. The recognition accuracy and precision are further improved.
The authors should highlight the area in figure 2 to demonstrate the features that the authors wanted the readers to see.
REPLY:
Thank you for your suggestion. The deviation in this section is due to the inconsistency between the original data position and the image display position. The deviation between the data position (dot) and the image display position (block diagram) can be intuitively felt through the image. For example, some data points may exceed the position of the block diagram, indicating numerous deviations in the image. This section is only for the display of deviations.
Please list the label of x axis in figure 10 and 11.
REPLY:
Thank you for your suggestion. The missing parts have been marked. The horizontal axis represents the proportion of the number of variables in the input test set (such as the number of business types of government agencies) to the total number of variables. The vertical axis represents the correlation between land type and variables (such as the correlation between A1 administrative office land and government agencies, and the correlation between R2 Class II residential land and plot ratio).
Please try to combine some figures with multiple sub panels to make the figures more informative. Please add some description in the figure caption so that reader can know what the authors wanted to show.
REPLY:
Thank you for your suggestion. We have summarized and presented multiple sets of data to enhance the readability of the paper. The details are as follows.
Currently, our method in recognition of land, the recognition of Class A and Class B land is ac-curate for small categories, the overall accuracy rate exceeds 85%. Classes R, S, and U land are accurate for the middle class, the overall accuracy rate reaches more than 94%. The above results indicate that the proposed method outperforms previous methods in the recognition accuracy of Classes R, S, and U land.
Round 2
Reviewer 2 Report
The authors have made substantial improvements to the manuscript in response to my comments. As a result, I believe it is now ready for publication and I recommend accepting it.
Minor editing of English language required
Reviewer 3 Report
The authors have given comprehensive response to the reviewer and the reviewer agreed to accept it.
Need to polish.